# Concerto on Chromatin: Interplays of Different Epigenetic Mechanisms in Plant Development and Environmental Adaptation

**DOI:** 10.3390/plants10122766

**Published:** 2021-12-14

**Authors:** Jiao Liu, Cheng Chang

**Affiliations:** College of Life Sciences, Qingdao University, Qingdao 266071, China; jiaoliu6111@163.com

**Keywords:** DNA methylation, histone modifications, chromatin remodeling, noncoding RNAs, plant epigenetics, epigenetic interplays

## Abstract

Epigenetic mechanisms such as DNA methylation, histone post-translational modifications, chromatin remodeling, and noncoding RNAs, play important roles in regulating plant gene expression, which is involved in various biological processes including plant development and stress responses. Increasing evidence reveals that these different epigenetic mechanisms are highly interconnected, thereby contributing to the complexity of transcriptional reprogramming in plant development processes and responses to environmental stresses. Here, we provide an overview of recent advances in understanding the epigenetic regulation of plant gene expression and highlight the crosstalk among different epigenetic mechanisms in making plant developmental and stress-responsive decisions. Structural, physical, transcriptional and metabolic bases for these epigenetic interplays are discussed.

## 1. Introduction

In the natural environment, plants sense and respond to developmental and environmental cues by dedicated signaling pathways, which ultimately leads to extensive reprogramming of the transcriptome essential to various biological processes, including plant growth, development, and adaptation to environments [1,2]. Increasing evidence has revealed that plants have evolved sophisticated regulatory mechanisms to tightly control this transcriptomic reprogramming [1,2]. Transcription factors usually function as major regulators governing the expression of their target genes in response to developmental and environmental cues [1,2]. In addition, epigenetic processes and elements such as DNA methylation, histone post-translational modifications, chromatin remodeling, and noncoding RNAs (ncRNAs) are also widely involved in the regulation of gene expression at the transcriptional level, and play key roles in modulating plant developmental processes and responses to environmental stresses [3,4,5,6,7,8]. Past decades have seen unprecedented progress in the understanding of the epigenetic regulation of plant gene expression, and has revealed that distinct epigenetic mechanisms are highly interconnected and usually cooperate to fine-tune the gene expression essential to plant development processes and responses to environmental stresses [5,6,7,8,9,10,11,12]. Here, we reviewed these epigenetic mechanisms in regulating plant gene expression and discussed their multifaceted interplay in making developmental and stress-responsive decisions.

## 2. Epigenetic Mechanisms Regulating Plant Gene Expression

As one of the most important epigenetic mechanisms controlling plant gene expression and genome stability, DNA methylation mainly refers to the 5-methylcytosine (5-mC) methylation which occurs by the addition of a methyl group to the fifth carbon position in the pyrimidine ring of cytosine in the DNA sequence context of symmetric CG and CHG and the asymmetric CHH (where H is any nucleotide except G) [13]. Recently, extensive methylation at the fourth nitrogen position of cytosine was detected in the genome of liverwort *Marchantia*
*polymorpha* and was demonstrated to be involved in *Marchantia* spermatogenesis [14]. In plants, the pattern of DNA methylation is shaped by de novo methylation, maintenance methylation, as well as demethylation. In the model plant *Arabidopsis thaliana*, de novo methylation is catalyzed by Domains Rearranged Methyltransferase 2 (DRM2) via the small RNAs (sRNAs)-dependent DNA methylation (RdDM) pathway [15,16]. Once established, DNA methylation in the context of CG, CHG, and CHH is maintained by METHYLTRANSFERASE 1 (MET1), plant-specific CHROMOMETHYLASES 2 (CMT2)/CMT3, and DRM2/CMT2, respectively [17,18]. As a reversible epigenetic mark, DNA cytosine methylation could be removed in passive and active demethylation pathways. Unlike passive demethylation mainly occuring during DNA replication, active demethylation is catalyzed by specific DNA glycosylases such as *Arabidopsis* DEMETER (DME) and REPRESSOR OF SILENCING 1 (ROS1), and play important roles in the regulation of gene expression [19,20,21]. DNA methylation could occur in many chromatin regions including genes and transposable elements (transposons) in model and crop plants. Enrichment of 5-mC in gene promoters usually represses gene transcription, whereas heavy methylation of transposons generally contributes to the transposon silencing and genome stability [22,23].

In plants, histone N-terminal tails hanging out of nucleosomes are usually subject to several types of post-translational modifications (PTMs) such as acetylation, methylation, phosphorylation, and ubiquitylation, which generally affect chromatin structure and gene transcription [24,25,26]. Through the addition of acetyl groups onto histone lysine residues such as H3K9/14/36 and H4K5/8/12/16, histone acetylation catalyzed by histone acetyltransferases (HATs) could relax chromatin structure and is generally associated with gene activation [27]. As reversible marks, these acetyl groups could be removed by histone deacetylases (HDACs) in the histone deacetylation process, which usually contributes to gene repression [27]. Another well-characterized histone PTM is histone methylation, which is dynamically regulated by histone methyltransferases and demethylases and is essential to the regulation of chromatin packaging and gene expression [28,29]. For instance, methylation of histone H3 lysine 4/36 (H3K4/36) and histone H3 lysine 9/27 (H3K9/27) generally contributes to gene activation and repression, respectively [28,29]. As an evolutionarily conserved, repressive epigenetic mark, histone H3 lysine 9 dimethylation (H3K9me2) generally contributes to the transcription repression [28,29] In addition, histone phosphorylation and ubiquitylation also play key roles in regulating gene expression [30]. Generally, histone phosphorylation at histone H3 serine 10/28 (H3S10/28) and histone H2A serine 95 (H2AS95), mediated by specific protein kinase, is generally associated with the gene activation, and histone phosphorylation at histone H3 theronine3 (H3T3) catalyzed by protein kinase MUT9p is associated with the gene silencing in *Chlamydomonas reinhardtii* [31]. H2B monoubiquitylation mediated by the H2B ubiquitin ligase could contribute to both transcription activation and elongation, whereas H2B demonoubiquitylation mediated by the ubH2B deubiquitinase is essential to the later stage of transcription elongation [32]. In contrast, H2A monoubiquitylation mediated by the H2A ubiquitin ligase is usually associated with transcriptional repression [33].

In eukaryotes, chromatin is compactly condensed and tightly coiled, limiting the accessibility of DNA for the transcription machinery and regulatory proteins. To initiate gene transcription, chromatin requires remodeling to open its compact structure and overcome the nucleosome barrier. ATP-dependent chromatin remodelers could utilize the energy of ATP hydrolysis to alter nucleosome distribution and/or composition to reorganize chromatin structure [34,35,36,37,38,39,40,41,42]. For instance, the deposition of H2A.Z, a variant form of canonical H2A, is mediated by the evolutionarily conserved SWR1 remodeling complex (SWR1c) [34,35]. In addition, some chromatin remodelers were reported to be involved in chromatin remodeling through mediating histone modifications [36,37]. Rice ATP-dependent chromodomain helicase-DNA binding 3 (CHD3)-type chromatin-remodeling factor CHR729, together with its ortholog in bread wheat, could regulate the methylation of H3K4 and H3K27, thereby regulating gene expression essential to plant development and secondary metabolite biosynthesis [38,39]. Increasing evidence has revealed that these ATP-dependent chromatin remodelers could act as both activators and repressors of gene expression in model and crop plants [40,41,42].

Noncoding RNAs (ncRNAs) that lack a protein-encoding function play key roles in the regulation of plant gene expression [43,44,45,46,47,48,49,50,51,52]. Based on their length, regulatory ncRNA could be separated into long ncRNAs (lncRNAs) and short ncRNAs such as microRNAs (miRNAs) and small interfering RNAs (siRNAs) [43,44,45,46]. Generally, lncRNAs contain more than 200 nucleotides (nts), whereas miRNA contain 18-24 nts. As highly heterogeneous molecules in biogenesis, lncRNAs could be transcribed from intergenic regions, introns and antisense strands of protein-coding genes by DNA-dependent RNA polymerase II (Pol II), III (Pol III), IV (Pol IV), and V (Pol V) in plants [47,48,49,50]. Nascent lncRNAs transcribed by Pol II require capping, splicing and polyadenylation to become mature, but Pol IV/V-dependant lncRNAs donot undergo these processing steps. miRNAs are firstly transcribed as primary miRNAs (pri-miRNAs) by Pol II from miRNA-encoding genes (MIRs). In the nucleus, nascent pri-miRNAs are preferentially processed by the RNase III enzymeDicer-Like1 (DCL1) into the precursor miRNAs (pre-miRNAs) as miRNA/miRNA* duplexes. After methylation by the nuclear HUA ENHANCER 1 (HEN1) protein, these miRNA/miRNA* duplexes are exported to the cytoplasm, where one strand of each duplex is recruited by the ARGONAUTE1 (AGO1) protein to form the miRNA-mediated silencing complex (miRISC) for mediating gene silencing [51,52].

## 3. Interplays of Different Epigenetic Marks Regulating Plant Gene Expression

Although different epigenetic marks such as DNA methylation, histone modifications, as well as histone variants, are deposited by distinct epigenetic modifiers, they usually function in concert rather than independently, to determine plant gene expression in plant development and stress adaptation. For instance, several epigenetic marks areshown to be colocalized to certain regions of the plant genome, and tend to co-occur at plant developmental stages and/or in responses to environmental stresses [9,10]. Increasing evidence has revealed that DNA methylation marks are usually associated with histone modification marks. Genome-wide profiling studies of DNA methylation and histone methylation through using whole-genome bisulfite sequencing (WGBS), chromatin immunoprecipitation (ChIP) coupled with microarray (ChIP-chip), and ChIP coupled with sequencing (ChIP-seq), reveal that levels of DNA methylation are highly associated with levels of dimethylation of H3K9 (H3K9m2) throughout the genome, especially at transposons, in the model plant *Arabidopsis* and the crop plants maize and *Brassica rapa* L. [53,54,55,56,57,58,59]. Decreased levels of DNA methylation were found together with increased levels of histone modification marks H4K5K8K12K16ac, H3K4me3, H3K4me2, and H3K36me2 at transposable elements (TEs) in the *Arabidopsis* mutant of histone deacetylase gene *HDA6*, suggesting that DNA methylation was modulated together with histone acetylation and methylation by *HDA6* to maintain the TE silencing [60]. Similarly, H3K9me2 hypermethylation and H3K27me3 redistribution were observed together with ectopic DNA methylation at PcG (Polycomb Group proteins) target genes in the *Arabidopsis* mutant of DNA methyltransferase gene *MET1* [60]. Notably, monomethylation of H3K23 (H3K23me1) was found to be preferentially associated with CG DNA methylation at gene bodies and pericentromeric regions in *A. thaliana* [61]. In addition, decreased levels of DNA methylation were identified together with increased levels of histone H2B monoubiquitination, and released silencing of transgenes in the mutant of *Arabidopsis* H2B deubiquitinase gene *SUP32*/*UBP26*, elucidating the association of DNA methylation with histone ubiquitination in maintaining plant gene silencing [62]. Therefore, DNA methylation marks are functionally associated with multiple histone modification marks such as histone methylation, acetylation and ubiquitination in model and crop plants.

Crosstalk among different histone modification marks was uncovered by genome-wide profiling of histone modifications in model and crop plants. For instance, genome-wide profiling of histone modifications H3K9ac, H3K27ac, H3K4me2, H3K4me3, H3K9me3, and H3K27me3 in the model plant *Arabidopsis* and the crop plant rice using ChIP-seq revealed that permissive epigenetic marks H3K9/K27ac likely co-occur with H3K4me2/3 at genome regions, whereas repressive mark H3K9me3 occurs concurrently with another repressive mark, H3K27me3 [58,59]. Interestingly, permissive epigenetic marks H3K4me2/3 were revealed to occur in conjunction with repressive mark H3K27me3 to define the bivalent epigenetic marks, which are widely involved in plant adaptation to environmental stresses. For instance, bivalent histone modification H3K27me3-H3K4me3 was found enriched in the *Arabidopsis* vernalization gene *VERNALIZATION INSENSITIVE 3* (*VIN3*) after induction by prolonged cold treatment [63]. Similarly, cold stress could induce the enrichment of the bivalent H3K4me3-H3K27me3 epigenetic mark at the gene body regions of active genes in potato, which might increase the gene accessibility and facilitate gene regulation in response to cold stress [63]. In addition, crosstalk among different histone modification marks was revealed by studies on the mutants of epigenetic modifiers. For instance, loss-of-function of the *Arabidopsis* H2B deubiquitinase gene *SUP32*/*UBP26* resulted in the decreased level of H3K9me2 but increased H2Bub enrichment at heterochromatic regions [62]. Similarly, increased levels of H4K5K8K12K16ac, H3K4me3, H3K4me2, and H3K36me2 were found at TEs in the *Arabidopsis*
*hda6* mutant [60].

Links between histone variants and histone modifications were revealed by recent studies in the model plant *Arabidopsis*. As main variant forms of canonical H2A, H2A.Z deposited by the evolutionarily conserved chromatin remodeling complex SWR1c play important roles in the regulation of chromatin structure essential to gene expression [64,65]. It was demonstrated that the *Arabidopsis* MUT9P-LIKE-KINASE (MLK4) could phosphorylate histone H2A on serine 95 (H2AS95). Interestingly CIRCADIAN CLOCK ASSOCIATED1 (CCA1) could interact with both MLK4 and the SWR1 co-subunit YAF9A [65]. Notably, the *Arabidopsis mlk4* mutant displayed late flowering, as well as reduced H2AS95 phosphorylation, H2A.Z accumulation, H4 acetylation, and the expression of *GI*, suggesting that histone variant H2A.Z functions in concert with H2AS95 phosphorylation and H4 acetylation, to regulate *GI* expression and flowering time in *Arabidopsis* [65].

In addition, the relationship between ncRNAs and epigenetic marks has been extensively characterized in the past decade, and has revealed that ncRNAs, especially lncRNAs, could modulate epigenetic marks to fine-tune gene expression in plant development and stress response. For instance, *COLD ASSISTED INTRONIC NONCODING RNA (COLDAIR)* is a group of antisense lncRNAs transcribed by Pol II from the *Arabidopsis*
*FLOWERING LOCUS C* (*FLC*) locus. It was demonstrated that *COOLAIR* could associate in ciswith the *FLC* locus and contribute to the *FLC* transcriptional shutdown during vernalization [66]. Interestingly, disruption of *COOLAIR* expression attenuated the cold-dependent removal of H3K36 methylation, without alteration in H3K27me3 accumulation dynamics at the intragenic *FLC* nucleation site during the cold treatment, suggesting that *Arabiodopsis* lncRNA *COLDAIR* mediates the reduction in levels of H3K36me3 at *FLOWERING LOCUS C* (*FLC*), thereby coordinating the epigenetic switching of chromatin state and promoting *FLC*-silencing during vernalization [66].

## 4. Molecular Basis for Epigenetic Interplays in Plant Development and Environmental Adaptation

### 4.1. Structural Basis for the Direct Links between Non-CG DNA Methylation and Histone H3K9me2

In plants, CG and CHG methylation are, respectively. maintained by MET1 and CMT3, whereas CHH methylation in the pericentromeric heterochromatin regions is maintained by CMT2. It is widely demonstrated that maintenance of CG DNA methylation does not rely on the presence of histone epigenetic mark H3K9me2, but non-CG DNA methylation (CHG and CHH methylation) maintenance requires histone H3K9me2. Structural and functional analyses of DNA methyltransferase CMT2, CMT3, histone H3K9 methyltransferase SUPPRESSOR OF VARIEGATION 3-9 HOMOLOG 4 (SUVH4) and SUVH6 provide novel insight into the molecular mechanisms underlyingdirect links between non-CG DNA methylation and histone H3K9me2 [67,68,69,70,71,72]. For instance, the structure analysis of Arabidopsis CMT3 and maize ZMET2 (a maize ortholog of *Arabidopsis* CMT3) revealed that both proteins contain bromo-adjacent homology domains (BAH domains) and chromodomains with H3K9me2 binding sites to target these enzymes to H3K9me2 containing peptides (Figure 1A) [67,68,69,70]. Similarly, another *Arabidopsis* DNA methyltransferase CMT2 was revealed to preferentially recognize the H3K9me2 mark *in vitro* and *in planta* (Figure 1A) [67,68,69,70]. In addition, crystal structure analysis of SUVH4 and SUVH6 revealed that their N-terminal SRA (SET- and RING-associated) domains could form a DNA binding cleft and establish charged interaction with methylated DNA, suggesting that methylated DNA might recruit SUVH family H3K9 methyltransferases to chromatin and promote histone H3K9 dimethylation (Figure 1B) [71,72]. Interestingly, in vitro and in vivo experiments revealed that the SUVH family H3K9 methyltransferases SUVH4, SUVH5 and SUVH6 have distinct binding preferences to context-biased non-CG DNA methylation, thereby targeting H3K9 methylation to sites with different methylated DNA sequences in plants [72].Taken together, these structural and biochemical analyses of DNA methyltransferase CMT2/3 and histone H3K9 methyltransferase SVUH4/6 shed novel light on the establishment of the reinforcing loop between the plant non-CG DNA methylation and histone H3K9 dimethylation.

### 4.2. Physical Basis for the Epigenetic Interplayes Mediated by Chromatin Modifier Complex

In plants, different epigenetic modifiers usually interact with each other and function in concert to regulate gene expression, which might underlie the complex interplay between different epigenetic marks in regulating plant gene expression. Indeed, some DNA methylation enzymes could directly interact with histone modification enzymes and/or ATP-dependent chromatin remodelers. For instance, *Arabidopsis* DNA methyltransferase MET1 was demonstrated to interact with histone deacetylase HDA6 and thus maintain silencing of heterochromatin regions and transposable elements (Figure 2A) [60,73,74]. Interestingly, McHDA6 could bind directly to McMET1 to suppress the anthocyanin biosynthesis in leaves of *Malus* crabapple under inorganic phosphorus (Pi) sufficiency, implying that HDA6-MET1 association might be conserved across plant species [75]. It is well known that DNA methylation has been employed by plants as an epigenetic defense mechanism to defend against geminivirus infection [76]. Interestingly, the V2 protein of Tomato yellow leaf curl virus (TYLCV) could interact with the *Nicotiana benthamiana* histone deacetylase 6 (NbHDA6) to interfere with the NbHDA6-NbMET1 association and inhibit the methylation of viral DNA genome, thereby compromising the host’s resistance against TYLCV infection [76]. At the same time, *Arabidopsis* DNA methyltransferase MET1 was also revealed to interact with MEDEA, a Polycomb group protein essential for the deposition of H3K27me3 marks, to repress seed development (Figure 1B) [77]. In addition, another DNA methylation component, SU(VAR)2, was demonstrated to interact with chromatin remodeler CHR19/27/28 to maintain transcriptional gene silencing in *A. thaliana*, elucidating the association of plant DNA methylation components with chromatin remodelers [78].

The association among different histone modification enzymes and/or ATP-dependent chromatin remodelers has been revealed by increasing evidence. For instance, the RPD3-type histone deacetylase HDA6 could interact with type-2 histone deacetylase AtHD2C to regulate gene expression in the ABA and salt-stress response in *Arabidopsis* [79,80]. TaHDA6, the wheat ortholog of *Arabidopsis* AtHDA6, could also interact with the type-2 histone deacetylases AtHDT701 to regulate expression of defense-related genes in bread wheat, suggesting that the physical and functional association of HDA6 with type-2 histone deacetylase might be conserved among monocots and dicots [81,82]. At the same time, HDA6 was revealed to interact with multiple histone modification enzymes and chromatin remodelers, including histone demethylase FLOWERING LOCUS D (FLD), histone methyltransferases SUVH4/5/6, SWI/SNF chromatin-remodeling complex component AtSWI3B and polycomb repressive complex 2 (PRC2) subunit MSI1, to regulate plant gene expression through multiple epigenetic mechanisms [83,84,85,86,87]. For instance, histone deacetylase HDA6 interacts with MET1, SUVH4/5/6, and SWI3B, to maintain transposon silencing by decreasing histone H3 acetylation and nuclear occupancy, as well as increasing DNA methylation and histone H3K9me2 (Figure 2B) [60,74,83,84]. Similarly, HDA6 also associates with FLD and MSI1 to repress the expression of *FLC*, *MAF4*, and *MAF5* via decreasing H3K9Ac and promoting H3K27Me3, thereby regulating the *Arabidopsis* flowering time (Figure 2B) [85,86]. In addition, the type-2 histone deacetylase HD2C also could interact with a BRM-containing SWI/SNF chromatin-remodeling complex to regulate expression of heat-responsive genes in *A. thaliana*, thereby controlling plant heat-stress response, but the regulation of histone deacetylation and chromatin remodeling at heat-responsive genes by the HD2C-SWI/SNF complex remains to be explored in future research [87]. Therefore, these studies support the fact that different epigenetic modifiers could associate with each other to regulate plant gene expression through coordinating different epigenetic mechanisms, thereby providing a physical basis for the epigenetic interplays in making plant developmental and stress-responsive decisions.

### 4.3. Trancriptional Basis for the Cross-Regulation of Different Epigenetic Mechanisms

In addition to physical association, different epigenetic modifiers could regulate each other at the transcriptional level, thereby contributing to the complex interplays of different epigenetic mechanisms in regulating plant gene expression. For instance, the proper expression of the *Arabidopsis* H3K9 demethylase gene *IBM1* (*Increase in BONSAI methylation 1*) relies on the CG and CHG methylation mediated by the DNA methyltransferase MET1 and the plant-specific chromomethylase CMT3 in a large intron of *IBM1* (Figure 3A) [88]. In addition, recent studies revealed that expressions of *ncRNA* genes were fine-tuned by DNA methylation, histone modification, and even histone variants. For instance, DNA methylation at flanking regions is negatively correlated with the expression of *MIR172b* genes in the bisexual flower development of andromonoecious poplar [89]. In addition, expressions of *Arabidopsis*
*MIR156a*, *MIR164a*, *MIR165a*, *MIR168a*, *MIR172a*, *MIR395e*, and *MIR399d* genes are positively regulated by GCN5-mediated histone H3K14 acetylation (Figure 3B) [90]. The *Arabidopsis gcn5* mutant displays a pleiotropic developmental phenotype such as curled leaves and altered floral organ numbers, which is accompanied by altered levels of *miR159*, *miR162*, *miR167* and *miR168*, implying that transcriptional regulation of these *MIR**NA* genes by GCN5-mediated histone acetylation contributes to development processes in *A. thaliana.* Transcriptional activation of *Arabidopsis*
*MIR156*, *MIR164*, and *MIR396* genes is associated with the H2A.Z deposition mediated by the SWR1 chromatin remodeling complex (SWR1-C) (Figure 3C) [91,92]. Expression of *Arabidopsis MIR156* and *MIR164* genes were attenuated in the mutants of SWR1-C components such as *arp6* and *pie1*, which is correlated with the activation of target development genes such as *SQUAMOSA PROMOTER BINDING PROTEIN-LIKE 3* (*SPL3*), *SPL4*, *SPL9*, *CUP-SHAPED COTYLEDON1* (*CUC1*) and *CUC2* [91]. Similarly, the spatiotemporal expression pattern of *Arabidopsis MIR396* gene was altered in the *arp6* and *pie1* mutants, leading to the affected expression of target development genes *CRY2-INTERACTING BHLH 4 (CIB4)* and *FLOWERING LOCUS T (FT)* [92]. These studies revealed that different epigenetic modifiers could regulate each other at the transcriptional level.

### 4.4. Metabolic Basis for the Crosstalk of DNA and Histone Methylation

Increasing evidence has revealed that crosstalks between DNA and histone methylation could occur at the metabolic level. For instance, the one-carbon metabolism contributes to both DNA and histone methylation by generating S-adenosylmethionine (SAM), the universal methyl group donor for the methylation of DNA and proteins. The *Arabidopsis* folylpolyglutamate synthetase FOLYLPOLYGLUTAMATE SYNTHETASE 1 (FPGS1) mediates the folate polyglutamylation to generate polyglutamated 5-methyl-tetrahydrofolate (5-CH3-THF-Glun), which could provide the methyl group for the production of methionine (Met), the precursor of SAM (Figure 4) [93]. The *fpgs1* mutation attenuated DNA methylation and histone H3K9 dimethylation in *A. thaliana*, which could be rescued by exogenous application of 5-formyltetrahydrofolate, suggesting that one-carbon metabolism directly contributes to the interplay of DNA and histone methylation [93]. Consistent with this, mutation of *Arabidopsis METHIONINE SYNTHASE1* (*ATMS1*), which encode cobalamin-independent Met synthases functioning in the one-carbon metabolism pathway, attenuated both DNA methylation and histone H3K9 dimethylation, leading to the release of chromatin silencing of endogenous genes and transposons (Figure 4) [94]. In the one-carbon metabolism cycle, metadenosyltransferase 4 (MAT4) catalyzes the SAM biosynthesis (Figure 4) [95]. It was recently demonstrated that *mat4* mutation decreased levels of DNA methylation and histone H3K9me2 in *Arabidopsis*, further confirming the importance of the one-carbon metabolism cycle in the interplay of DNA and histone methylation [95]. Although the *mat1mat2* double mutant showed normal growth and fertility, the *mat1mat4* double mutant is smaller than the wild type. In addition, the *mat2mat4* double mutant has embryonic defects, suggesting that MAT4 plays a predominant role in plant growth and development in *Arabidopsis* [95]. These studies suggest that the one-carbon metabolism contributes to both DNA and histone methylation essential to the plant development processes.

## 5. Concluding Remarks and Perspectives

In this review, we provided an overview ofrecent proceedings in understanding the regulationof plant gene expression by multiple epigenetic mechanisms, and highlighted the cross-regulation of these epigenetic mechanisms in governing gene expression essential toplant development and stress responses. In addition, we discussed the structural, physical, transcriptional and metabolic basis for the epigenetic interplays that greatly contribute to the complexity of transcriptional reprogramming in plant development processes and responses to environmental stresses. Although recent decades have seen great progress in understanding the epigenetic interplay in controlling plant gene expression, we still have a long way to go towards fully understanding the complicated interplay of different epigenetic mechanisms in regulating plant development and stress responses. For instance, structural and biochemical analysis of *Arabidopsis* DNA methyltransferases CMT2/3 and histone H3K9 methyltransferasesSUVH4/6 have provided insight into cross-regulation of DNA and histone methylation by CMT2/3 and SUVH4/6, but structures of other chromatin modifiers and their contributions to the epigenetic interplays remains to be disclosed. Furthermore, identification of interactors with *Arabidopsis* HDACs such as AtHDA6 shed novel light onto the crosstalk of histone deacetylation with other types of chromatin modifications, and characterizing interacting proteins of other epigenetic modifiers would provide more insight into the cross-regulation of different epigenetic mechanisms. In addition, the one-carbon metabolism pathway was widely demonstrated to regulate DNA and histone methylation, but involvement of other metabolism pathways in the epigenetic interplays remains to be explored in future research.

## Figures and Tables

**Figure 1 plants-10-02766-f001:**
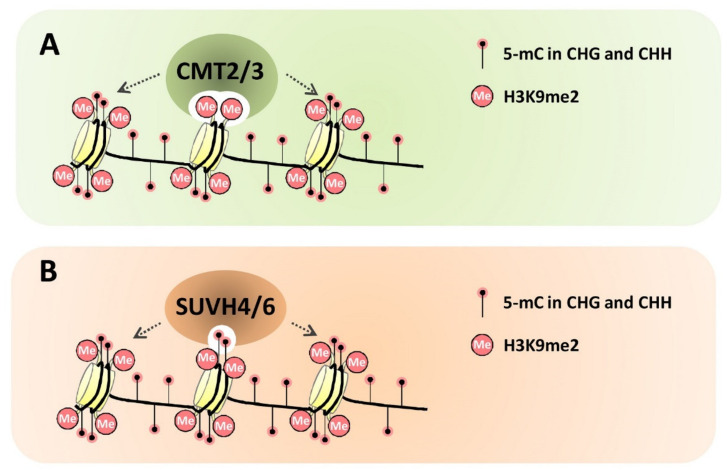
Simplified models for the DNA and histone methylation mediated by DNA methyltransferases CMT2/3 and histone H3K9 methyltransferases SUVH4/6. (**A**) *Arabidopsis* DNA methyltransferases CMT2/3 contain H3K9me2 binding sites and directly link non-CG DNA methylation and histone H3K9me2. (**B**) *Arabidopsis* histone H3K9 methyltransferases SUVH4/6 contain methylated DNA binding sites, thereby mediating the cross-regulation of non-CG DNA methylation and histone H3K9me2.

**Figure 2 plants-10-02766-f002:**
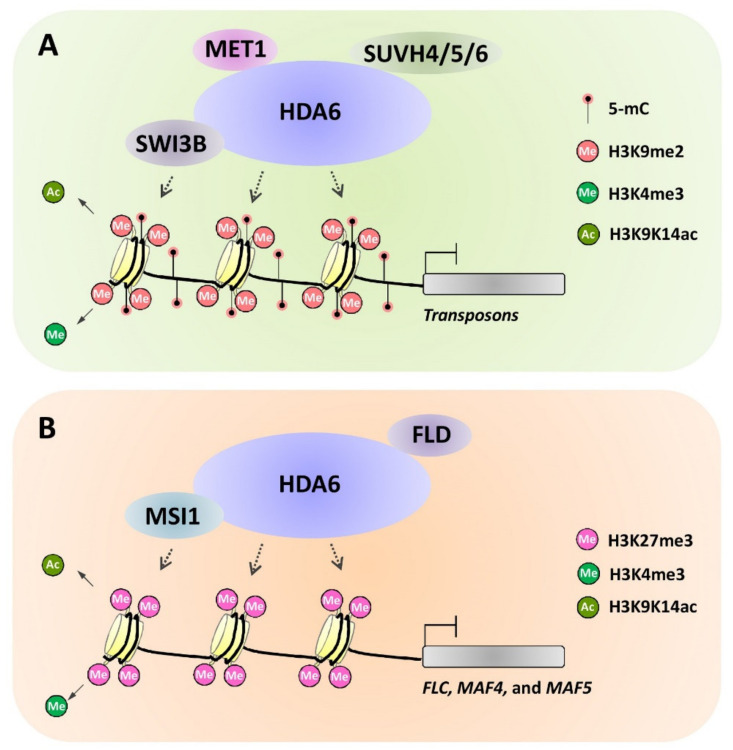
Simplified models for the epigenetic interplays mediated by HDA6 complex. (**A**) HDA6 interacts with MET1, SUVH4/5/6, and SWI3B, to maintain the transposon silencing by decreasing histone H3 acetylation and nuclear occupancy, as well as increasing DNA methylation and histone H3K9me2 in *A. thaliana*. (**B**) Histone deacetylase HDA6 associates with histone demethylase FLD and polycomb repressive complex 2 (PRC2) subunit AtMSI1 to repress the expression of *FLC*, *MAF4*, and *MAF5* via decreasing H3K9Ac and promoting H3K27Me3, thereby fine-tuning the *Arabidopsis* flowering time.

**Figure 3 plants-10-02766-f003:**
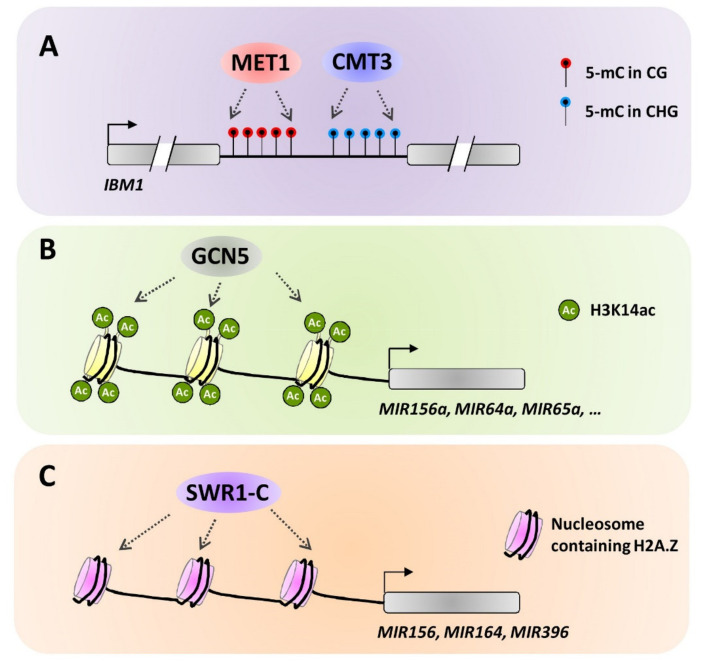
Simplified models for the transcriptional mechanisms of epigenetic interplays. (**A**) *Arabidopsis* DNA methyltransferase MET1 and plant-specific chromomethylase CMT3 directly regulate expression of the H3K9 demethylase gene *IBM1* by mediating CG and CHG methylation in a large intron of *IBM1*. (**B**) GCN5-mediated histone H3K14 acetylation positively regulates expressions of *MIR156a*, *MIR164a*, *MIR165a*, *MIR168a*, *MIR172a*, *MIR395e*, and *MIR399d* genes in *A. thaliana*. (**C**) SWR1 chromatin remodeling complex (SWR1-C)-mediated H2A.Z deposition activates expression of *MIR156*, *MIR164*, and *MIR396* genes in *A. thaliana*.

**Figure 4 plants-10-02766-f004:**
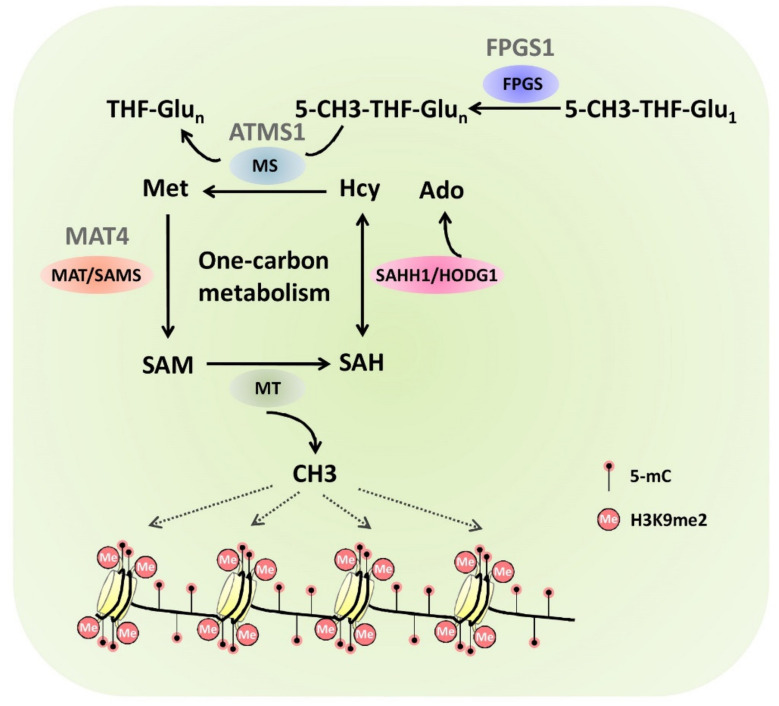
Simplified models for one-carbon metabolism pathway contributing to the DNA and histone methylation by producing the methyl group donor SAM.

## Data Availability

Not applicable.

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
