# Peer review of "Concerto on Chromatin: Interplays of Different Epigenetic Mechanisms in Plant Development and Environmental Adaptation"

_plants, 2021, doi:10.3390/plants10122766_

Round 1

Reviewer 1 Report

This review, entitled "Concerto on the chromatin: interplays of different epigenetic mechanisms in plant development and environmental adaptation" provides a broad overview of the current knowledge on the crosstalk between different epigenetic mechanisms, as related to plant development and stress adaptation. However, some problems in the manuscript worth mentioning and should be resolved.

  • In the Abstract, please omit “even” (page 1, line 10) - it is not clear why even, the same is true for “these” (page 1, line 14) because the authors clarity the text further.
  • The “interplays” is not the best keyword, maybe “epigenetic interplay” would better reflect the research content of the manuscript.
  • page 1, lines 33-35: Rephrase the sentence “As well-studied epigenetic marks, DNA methylation and histone modifications could be sustained by mitosis and/or meiosis and contribute to both gene activation and silencing in plant development and stress response”. The first part of this sentence does not correspond to the second one, moreover, I would not say that DNA methylation and histone modifications are equally well studied.    
  • page 1, lines 35-38: “Through governing chromatin structure …… chromatin remodeling plays…” - this statement is not very logical and needs revision.
  • page 1, lines 38-41: Could you clarify “In addition, ncRNAs like long- noncoding RNAs (lncRNAs), small interfering RNA (siRNA), and microRNAs (miRNAs) could regulate gene expression through epigenetic mechanisms such as DNA methylation, histone modification, and even chromatin structure changes”, and be more specific by listing some concrete examples.
  • In the Introduction “plant development and stress response” is repeated 4 times, including twice in the same sentence (lines 43-45). Such repetitions must be avoided, and once again - try to be more specific.
  • page 2: Rephrase subtitle 2 “Epigenetic mechanisms and their regulation on gene transcription” – what do you mean?
  • page 2 and throughout the text: Latin names of the species must be always italicized, e.g. Marchantia polymorpha, Marchantia spermatogenesis, Arabidopsis thaliana, Chlamydomonas reinhardtii, Brassica rapa, Malus crabapple, etc. The same is true for gene symbols and mutant names throughout the manuscript text.
  • When you are talking about something that is known, check the correct use of the verb tenses, e.g. the past tense (lines 57 and 59) should be replaced with the present tense.
  • Page 2, line 70: instead of “nucleosomes usually subject”, use “nucleosomes ARE usually subject”
  • Page 3, line 134: Introduce the abbreviation of “WGBS”
  • Page 3, lines 146-147: Something is wrong with the sentence “..monomethylation of H3K23 (H3K23me1) WAS REVEALED ASSOCIATED with CG DNA methylation at gene bodies and pericentromeric regions in A. thaliana”, please correct.
  • The final part “Concluding Remarks and Perspectives” seems to be redundant and needs revision. Some of the statements actually overlap with the previous sections. Can the authors try to cut it short and highlight the main points and conclusions ?

Overall, this review manuscript provides a good analysis of the previously published works on the crosstalk between different epigenetic regulators in model and crop plants. It should be considered for publication in Plants, once all the issues indicated in the comments are solved.

Author Response

Review 1 # Comments and Suggestions for Authors

This review, entitled "Concerto on the chromatin: interplays of different epigenetic mechanisms in plant development and environmental adaptation" provides a broad overview of the current knowledge on the crosstalk between different epigenetic mechanisms, as related to plant development and stress adaptation. However, some problems in the manuscript worth mentioning and should be resolved.

- Response: Thank you very much for these comments! We fully agree with the Reviewer and have made extensive revision of this manuscript according to referee’s comments.

In the Abstract, please omit “even” (page 1, line 10) - it is not clear why even, the same is true for “these” (page 1, line 14) because the authors clarity the text further.

- Response: Many thanks. “even” has been removed from the new version of the Abstract.

The “interplays” is not the best keyword, maybe “epigenetic interplay” would better reflect the research content of the manuscript.

- Response: We thank Reviewer for the suggestion. “epigenetic interplay” instead of  “interplays” have been listed among keywords in the revised manuscript.

page 1, lines 33-35: Rephrase the sentence “As well-studied epigenetic marks, DNA methylation and histone modifications could be sustained by mitosis and/or meiosis and contribute to both gene activation and silencing in plant development and stress response”. The first part of this sentence does not correspond to the second one, moreover, I would not say that DNA methylation and histone modifications are equally well studied.

- Response: Thank you very much for identifying this mistake. This incorrect sentence has been removed in the revised manuscript.

page 1, lines 35-38: “Through governing chromatin structure …… chromatin remodeling plays…” - this statement is not very logical and needs revision.

- Response: Many thanks. This confusing sentence has been deleted in the revised manuscript.

page 1, lines 38-41: Could you clarify “In addition, ncRNAs like long- noncoding RNAs (lncRNAs), small interfering RNA (siRNA), and microRNAs (miRNAs) could regulate gene expression through epigenetic mechanisms such as DNA methylation, histone modification, and even chromatin structure changes”, and be more specific by listing some concrete examples.

- Response: We thank Reviewer for the suggestion. This confusing sentence has been removed in the revised manuscript.

In the Introduction “plant development and stress response” is repeated 4 times, including twice in the same sentence (lines 43-45). Such repetitions must be avoided, and once again - try to be more specific.

- Response: Thank you very much! We have rewritten these sentences and replaced these repetitions with specific examples in the revised manuscript.

page 2: Rephrase subtitle 2 “Epigenetic mechanisms and their regulation on gene transcription” – what do you mean?

- Response: Many thanks. Subtitle 2 has been rephrased as “Epigenetic mechanisms regulating plant gene expression” in the revised manuscript.

page 2 and throughout the text: Latin names of the species must be always italicized, e.g. Marchantia polymorpha, Marchantia spermatogenesis, Arabidopsis thaliana, Chlamydomonas reinhardtii, Brassica rapa, Malus crabapple, etc. The same is true for gene symbols and mutant names throughout the manuscript text.

- Response: We thank the Reviewer for identifying these mistakes and they are all corrected in the revised revision.

When you are talking about something that is known, check the correct use of the verb tenses, e.g. the past tense (lines 57 and 59) should be replaced with the present tense.

- Response: Many thanks. Present tense is used to describe a fact that has been already demonstrated and accepted by the scientific community in the revised manuscript.

Page 2, line 70: instead of “nucleosomes usually subject”, use “nucleosomes ARE usually subject”

- Response: We thank the Reviewer for identifying this mistake and it has been corrected as “nucleosomes are usually subject” in the new version.

Page 3, line 134: Introduce the abbreviation of “WGBS”

- Response: Yes, the abbreviation of “WGBS”has been introduced in the revised manuscript.

Page 3, lines 146-147: Something is wrong with the sentence “..monomethylation of H3K23 (H3K23me1) WAS REVEALED ASSOCIATED with CG DNA methylation at gene bodies and pericentromeric regions in A. thaliana”, please correct.

- Response: We thank the Reviewer for identifying this mistakes and it has been corrected in the revised revision

The final part “Concluding Remarks and Perspectives” seems to be redundant and needs revision. Some of the statements actually overlap with the previous sections. Can the authors try to cut it short and highlight the main points and conclusions ?

- Response: We are grateful for this suggestion. We have now rewritten the discussion section, deleted the redundant parts and and highlighted the main points and conclusions in the revised version. Hopefully, this version could meet the standard for publication.

Overall, this review manuscript provides a good analysis of the previously published works on the crosstalk between different epigenetic regulators in model and crop plants. It should be considered for publication in Plants, once all the issues indicated in the comments are solved.

- Response: Thank you very much for these encouraging comments and very helpful suggestions on this manuscript.

Reviewer 2 Report

This review focuses on the regulation of plant gene expression by multiple epigenetic mechanisms and discusses the cross-talk among these epigenetic mechanisms in plant development and stress responses. Along the review, the authors explain how the different epigenetic marks function in concert rather independently to regulate gene expression.

In general, the review covers the main aspects of the topic under discussion and highlights the recent advances made to get more knowledge on the interplay of the different epigenetic mechanisms in plant development and adaptation. The authors give a thorough and well documented information about the state of the art of the subject.

As an overall recommendation, this review is considered to be accepted in present form.

Author Response

Review 2 # Comments and Suggestions for Authors

This review focuses on the regulation of plant gene expression by multiple epigenetic mechanisms and discusses the cross-talk among these epigenetic mechanisms in plant development and stress responses. Along the review, the authors explain how the different epigenetic marks function in concert rather independently to regulate gene expression.

In general, the review covers the main aspects of the topic under discussion and highlights the recent advances made to get more knowledge on the interplay of the different epigenetic mechanisms in plant development and adaptation. The authors give a thorough and well documented information about the state of the art of the subject.

As an overall recommendation, this review is considered to be accepted in present form.

- Response: Thank you very much for these encouraging comments. We have made extensive revision of this manuscript according to referees’ comments. Hopefully, this version could meet the standard for publication.

Reviewer 3 Report

The work is not a very successful compendium of current knowledge on the subject. To be honest, I did not learn anything new from it - the fact that I am a bit stuck on the subject. There are quite a few better works on the subject and I see no point in publishing another.

Author Response

Review 3 # Comments and Suggestions for Authors

The work is not a very successful compendium of current knowledge on the subject. To be honest, I did not learn anything new from it - the fact that I am a bit stuck on the subject. There are quite a few better works on the subject and I see no point in publishing another.

- Response: Thank you very much for these comments. In contrast with other works, our paper provides a broad overview of the current knowledge on the crosstalk between different epigenetic mechanisms, as related to plant development and stress adaptation. In this comprehensive review, we explain how the different epigenetic marks function in concert rather independently to regulate gene expression. Furthermore, we highlighted the recent advances made to get more knowledge on the interplay of the different epigenetic mechanisms in plant development and adaptation. In addition, we have made extensive revision of this manuscript including figures according to referees’ comments and made a profound reorganization and gave a better description of different molecular details including concepts, pathways and mechanisms. Hopefully, this version could meet the standard for publication.

Reviewer 4 Report

The review presented by Liu and Chang focuses in the interplay between different epigenetic mechanisms in plant development and environmental adaptation. It contains an introduction based in the main epigenetic marks in plants, DNA methylation, histones modifications, chromatin remodellers and non-coding RNAs (ncRNAs) and three sections whose key messages is not clearly defined as their rationale is intertwined, specially between section 3 and 4. Some ideas and information are repeated throughout the manuscript with a mixture of concepts and topics in consecutive paragraphs. In my opinon, there is an excess of simplicity and lack of scientific rigor in describing some molecular mechanisms and pathways and this does not allow to get sensible information about the reviewed subject nor an idea of the latest findings in the field of epigenetics and plant development or environmental adaptation. Moreover, the relation between epigenetics mechanisms and plant development or environmental adaption is not obviously described on sections 3 and 4. Therefore, I believe the manuscript presented by the authors does not match with the given tittle.  The review needs a profound reorganization and a better description of the concepts and pathways. Furthermore, the writing of the manuscript should be improved as there are grammatical errors and repetitions. Consequently, this review is not suitable for publication in Plants. These are some of the concerns/observations that support my decision:

1.- Some statements made by the authors not precise or scientifically accurate. Here are just some examples:

- line 27-28: what do you mean by “cellular signal”?

- lines 38-41: the functions and biological implications of miRNA, siRNA and long noncoding RNAs are very different and quite complex. Authors simplify them so much, that the sentence in lines 38-41 is completely misleading to the reader.

- line 52: “base” is not correct. Should be “nucleotide”

- lines 61-62: “DNA demethylases in passive and active demethylation”. Wrong concept. It is just active demethylation

- line 64: DEMETER not DMEMER

- lines 68-69: the role of gene body methylation remains controversial. Sentence is not precise

- lines 80-81: H3K9me should be mentioned in the sentence as it is an evolutionary conserved repressive epigenetic mark.

- lines 92-93: “At the default state, chromatin composed of nucleosomes assembled from histones and DNAs is compactly condensed and tightly coiled, limiting the accessibility of DNA for transcription machinery and regulatory proteins.” Not scientifically accurate

- throughout the text the nomenclature used to name the histone modifications keeps changing and it is not scientifically accurate. For example: H3H4Me3 (line 140), H3K9m2 (line136), H3K23me1 (line 146)

- line 116: Not scientifically accurate. RNA PolV is not mentioned when explaining the RdDM. The description of RdDM is not scientifically accurate

- paragraph 106-119 is dreadful. miRNA formation and siRNA biogenesis are different in plants and animals. Authors mixed here both pathways making the whole process not comprehensive and with many conceptual errors.

- paragraph 120-123 is shocking. It is a long sentence but not clear or relevant information is given about lncRNAs

- in general section 3 seems a bunch of ideas not connected. Authors give a list of histone modifications with the enzymes involved on those modifications and keep jumping from one to another, making it really difficult to obtain a clear message from the text (specially lines 138-149). The connection with “plant development and environmental adaptation” is very weak in the section with just a couple of examples.

- section 3 starts with histone variants but they were not mentioned in the introduction as an epigenetic mark.

- lines 174-191: histone variants and ncRNAs are mixed in this paragraph. What is the rationale after this?

- what is the difference between section 3 and 4? They both seem to me the same thing and they really do not describe the connection between chromatin and plant development or environmental stress.

- line 285-286: not so simple. What happens with H3K4me for example?

- figures are really confusing and should be separated in independent figures as they show different processes. In general, figures are too small to show the molecular details (as the authors want to do in figure 1A) and too confusing as they mix different concepts, pathways and mechanism in the same figure (for example in figure 1B or 1C)

2.- Information and concepts are mixed and repetitive. Some sentences are too long and difficult to understand.

-the words “developmental processes and stress responses or developmental processes and environmental stresses” are repeated at least five times in the introduction (26 lines)

- the idea described in lines 38-41 of the introduction is again repeated in lines 106-115 of section 2. Moreover, there is a conceptual error as miRNA does not lead to DNA methylation (just in a couple of exceptions) as it is said in line 110.

- the authors abuse of the word “interplay” throughout the review. For example, between lines 192-224 is used four times

3.- Errors in grammar and style. Here are just some examples:

- in general authors should use the present tense when they are referring to a fact that has been already demonstrated and accepted by the scientific community. Many times they used past tense (example: line 127, “are” instead of “were”)

- line 28: to THE transcriptional machinery

- line 36: to THE transcriptional machinery

- line 37: plays AN important role

- line 70: nucleosomes ARE usually subjectED

- line 80: gets involved: out of context here

- line 82: As another two important… : not English

- line 92: chromatin composed of nucleosomes assembled from histones: not English

- line 129 and line 133: For instance (repetitive)

- line 138, and THE crop plants,

- line 141: in THE Arabidopsis mutant

- line 168: More crosstalk among… (not English)

Author Response

Review 4 # Comments and Suggestions for Authors

The review presented by Liu and Chang focuses in the interplay between different epigenetic mechanisms in plant development and environmental adaptation. It contains an introduction based in the main epigenetic marks in plants, DNA methylation, histones modifications, chromatin remodellers and non-coding RNAs (ncRNAs) and three sections whose key messages is not clearly defined as their rationale is intertwined, specially between section 3 and 4. Some ideas and information are repeated throughout the manuscript with a mixture of concepts and topics in consecutive paragraphs. In my opinon, there is an excess of simplicity and lack of scientific rigor in describing some molecular mechanisms and pathways and this does not allow to get sensible information about the reviewed subject nor an idea of the latest findings in the field of epigenetics and plant development or environmental adaptation. Moreover, the relation between epigenetics mechanisms and plant development or environmental adaption is not obviously described on sections 3 and 4. Therefore, I believe the manuscript presented by the authors does not match with the given tittle.  The review needs a profound reorganization and a better description of the concepts and pathways. Furthermore, the writing of the manuscript should be improved as there are grammatical errors and repetitions. Consequently, this review is not suitable for publication in Plants. These are some of the concerns/observations that support my decision:

- Response: Thank you very much for these comments! We fully agree with the Reviewer and have made extensive revision of this manuscript according to referee’s comments. In the revised manuscript, we made a profound reorganization and gave a better description of the concepts and pathways. Any inaccurate statement was removed from the revised version. Furthermore, we performed extensive English revision to avoid grammatical errors and repetitions in the revised manuscript. Moreover, we have incorporated the latest findings in the field of epigenetic regulation of plant development or environmental adaptation. In addition, figure 1 was separated in four figures to present different molecular details in a more readable way in the revised version. Hopefully, this version could meet the standard for publication.

  1. Some statements made by the authors not precise or scientifically accurate. Here are just some examples:

- line 27-28: what do you mean by “cellular signal”?

- Response: Thank you very much for identifying this mistake. This confusing expression has been rephrased in the revised manuscript.

lines 38-41: the functions and biological implications of miRNA, siRNA and long noncoding RNAs are very different and quite complex. Authors simplify them so much, that the sentence in lines 38-41 is completely misleading to the reader.

- Response: We thank Reviewer for the suggestion. These simplified and misleading descriptions about the functions and biological implications of miRNA, siRNA and long noncoding RNAs have been removed in the revised manuscript.

line 52: “base” is not correct. Should be “nucleotide”

- Response: Many thanks. “nucleotide” instead of “base” have been used in the revised sentence.

lines 61-62: “DNA demethylases in passive and active demethylation”. Wrong concept. It is just active demethylation

- Response: Thank you very much. This misleading description has been corrected in the new version.

line 64: DEMETER not DMEMER

- Response: Yes, we have corrected this typo in the revised version.

lines 68-69: the role of gene body methylation remains controversial. Sentence is not precise

- Response: Many thanks. This incorrect sentence has been removed from the revised manuscript.

lines 80-81: H3K9me should be mentioned in the sentence as it is an evolutionary conserved repressive epigenetic mark.

- Response: The reviewer is correct. We have mentioned H3K9me in the revised version.

lines 92-93: “At the default state, chromatin composed of nucleosomes assembled from histones and DNAs is compactly condensed and tightly coiled, limiting the accessibility of DNA for transcription machinery and regulatory proteins.” Not scientifically accurate

- Response: Thank you very much for identifying this mistake. This confusing sentence has been rephrased in the revised manuscript.

throughout the text the nomenclature used to name the histone modifications keeps changing and it is not scientifically accurate. For example: H3H4Me3 (line 140), H3K9m2 (line136), H3K23me1 (line 146)

- Response: Many thanks. These inaccurate nomenclature has been corrected in the revised manuscript.

line 116: Not scientifically accurate. RNA PolV is not mentioned when explaining the RdDM. The description of RdDM is not scientifically accurate

- Response: Thank you very much! We have rewritten this paragraph and removed these misleading sentences in the new version.

paragraph 106-119 is dreadful. miRNA formation and siRNA biogenesis are different in plants and animals. Authors mixed here both pathways making the whole process not comprehensive and with many conceptual errors.

- Response: Many thanks. We have rewritten this paragraph and removed these confusing sentences in the new version.

paragraph 120-123 is shocking. It is a long sentence but not clear or relevant information is given about lncRNAs

- Response: Thank you very much for identifying this mistake. This confusing sentence has been removed from the revised manuscript.

in general section 3 seems a bunch of ideas not connected. Authors give a list of histone modifications with the enzymes involved on those modifications and keep jumping from one to another, making it really difficult to obtain a clear message from the text (specially lines 138-149). The connection with “plant development and environmental adaptation” is very weak in the section with just a couple of examples.

- Response: We fully agree with the Reviewer and have rewritten section 3 in the revised manuscript.

section 3 starts with histone variants but they were not mentioned in the introduction as an epigenetic mark.

- Response: Many thanks. Histone variant H2A.Z is introduced a in the revised section 2.

lines 174-191: histone variants and ncRNAs are mixed in this paragraph. What is the rationale after this?

- Response: We thank the Reviewer for identifying this mistake. histone variants and ncRNAs were separated in the revised manuscript.

what is the difference between section 3 and 4? They both seem to me the same thing and they really do not describe the connection between chromatin and plant development or environmental stress.

- Response: We fully agree with the Reviewer and have rewritten section 3 and 4. In the revised section 3, interplays of epigenetic marks were introduced. Furthermore, we systematically discussed molecular basis for epigenetic interplays in plant development and environmental adaptation in the revised section 4. In addition, more connection between chromatin and plant development or environmental stress were introduced in the revised section 4.

line 285-286: not so simple. What happens with H3K4me for example?

- Response: Thank you very much for identifying this mistake. This confusing sentence has been rephrased in the revised manuscript.

figures are really confusing and should be separated in independent figures as they show different processes. In general, figures are too small to show the molecular details (as the authors want to do in figure 1A) and too confusing as they mix different concepts, pathways and mechanism in the same figure (for example in figure 1B or 1C)

- Response: We are grateful for Reviewer’s suggestions. Figure has been separated in four figures to present different molecular details in a more readable way in the revised manuscript.

  1. Information and concepts are mixed and repetitive. Some sentences are too long and difficult to understand.

-the words “developmental processes and stress responses or developmental processes and environmental stresses” are repeated at least five times in the introduction (26 lines)

- Response: Thank you very much. Repetitions of “developmental processes and stress responses or developmental processes and environmental stresses” have been removed from the revised introduction.

the idea described in lines 38-41 of the introduction is again repeated in lines 106-115 of section 2. Moreover, there is a conceptual error as miRNA does not lead to DNA methylation (just in a couple of exceptions) as it is said in line 110.

- Response: We thank the Reviewer for identifying these mistakes and they are all corrected in the revised revision.

the authors abuse of the word “interplay” throughout the review. For example, between lines 192-224 is used four times

- Response: We fully agree with the Reviewer. Repetitions of “interplay” were avoided in the new version.

  1. Errors in grammar and style. Here are just some examples:

- in general authors should use the present tense when they are referring to a fact that has been already demonstrated and accepted by the scientific community. Many times they used past tense (example: line 127, “are” instead of “were”)

- Response: We thank the Reviewer for identifying these mistakes. Present tense is used to describe something that is known in the revised manuscript.

line 28: to THE transcriptional machinery

- Response: Yes, this typo has been corrected in the revised version.

line 36: to THE transcriptional machinery

- Response: Yes, we have corrected this typo in the revised version.

line 37: plays AN important role

- Response: Many thanks. This typo has been corrected in the revised version.

line 70: nucleosomes ARE usually subjected

- Response: We thank the Reviewer for identifying these typos and they are all corrected in the revised revision.

- line 80: gets involved: out of context here

- Response: Thank you very much. We have corrected this mistake in the revised version.

line 82: As another two important… : not English

- Response: Many thanks. This incorrect expression has been rephrased in the revised version.

line 92: chromatin composed of nucleosomes assembled from histones: not English

- Response: We fully agree with the Reviewer. We have rewritten this sentence in the revised manuscript.

line 129 and line 133: For instance (repetitive)

- Response: Thank you very much! We have now rewritten this sentence and deleted these repetitions in the revised version.

line 138, and THE crop plants,

- Response: Yes, we have corrected this typo in the revised version.

line 141: in THE Arabidopsis mutant

- Response: Many thanks. We have corrected this mistake in the revised version.

line 168: More crosstalk among… (not English)

- Response: We thank the Reviewer for identifying these mistakes and they are all corrected in the revised revision.

Round 2

Reviewer 3 Report

In the current version, the work is acceptable.